# Michelangelo: Conditional 3D Shape Generation based on Shape-Image-Text Aligned Latent Representation

**Zibo Zhao**[1,2*]  **Wen Liu**[2*]  **Xin Chen**[2]  **Xianfang Zeng**[2]
**Rui Wang**[2]  **Pei Cheng**[2]  **Bin Fu**[2]  **Tao Chen**[3]
**Gang Yu**[2]  **Shenghua Gao**[1,4,5†]

[1]ShanghaiTech University    [2]Tencent PCG, China
[3]School of Information Science and Technology, Fudan University, China
[4]Shanghai Engineering Research Center of Intelligent Vision and Imaging
[5]Shanghai Engineering Research Center of Energy Efficient and Custom AI IC

https://github.com/NeuralCarver/Michelangelo

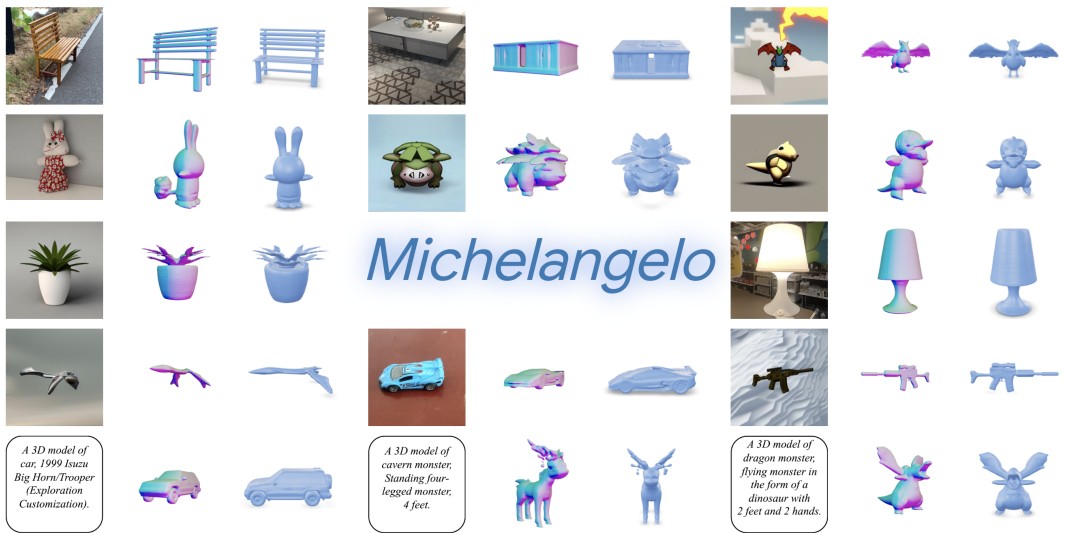

Figure 1: Visualization of the 3D shape produced by our framework, which splits into triplets with a conditional input on the left, a normal map in the middle, and a triangle mesh on the right. The generated 3D shapes semantically conform to the visual or textural conditional inputs.

## Abstract

We present a novel *alignment-before-generation* approach to tackle the challenging task of generating general 3D shapes based on 2D images or texts. Directly learning a conditional generative model from images or texts to 3D shapes is prone to produce inconsistent results with the conditions because 3D shapes have an additional dimension whose distribution significantly differs from that of 2D images and texts. To bridge the domain gap among the three modalities and facilitate multi-modal-conditioned 3D shape generation, we explore representing 3D shapes in a shape-image-text-aligned space. Our framework comprises two models: a Shape-Image-Text-Aligned Variational Auto-Encoder (SITA-VAE) and

---

*Contributed equally and work done while Zibo Zhao was a Research Intern with Tencent PCG.
†Corresponding author.

37th Conference on Neural Information Processing Systems (NeurIPS 2023).

a conditional Aligned Shape Latent Diffusion Model (ASLDM). The former model encodes the 3D shapes into the shape latent space aligned to the image and text and reconstructs the fine-grained 3D neural fields corresponding to given shape embeddings via the transformer-based decoder. The latter model learns a probabilistic mapping function from the image or text space to the latent shape space. Extensive experiments demonstrate that our proposed approach can generate higher-quality and more diverse 3D shapes that better semantically conform to the visual or textural conditional inputs, validating the effectiveness of the shape-image-text-aligned space for cross-modality 3D shape generation.

# 1 Introduction

Conditional generative model-based 3D shape generations, such as GAN [9, 35, 65], VAE [8, 36, 6], Auto-Regressive model [68, 39, 70], and Diffusion-based model [69, 41, 15, 13, 28, 42, 25], have great potential to increase productivity in the asset design of games, AR/VR, film production, the furniture industry, manufacturing, and architecture construction. However, two obstacles limit their ability to produce high-quality and diverse 3D shapes conforming to the conditional inputs: 1) diverse 3D shape topologies are complicated to process into a neural network-friendly representation; 2) since generating a high-quality 3D shape from a 2D image or textual description is an ill-pose problem, and also the distribution between the 3D shape space and image or text space is quite different, it is hard to learn a probabilistic mapping function from the image or text to 3D shape.

Recently, the neural fields in terms of occupancy [37, 45], Signed Distance Function (SDF) [43], and radiance field [38] have been driving the 3D shape representation in the computer vision and graphics community because their topology-free data structure, such as global latent [43], regular grid latent [45, 14], and point latent [69, 70], are easier to process for neural networks in an implicit functional manner. Once arrive at a compatible space to represent different topological 3D shapes, in light of the great success of auto-regressive and diffusion-based models in audio [26, 27], image [50, 51, 49, 54, 4], video [63, 59, 19, 7], and 3D human motions [72, 61, 66], a conditional auto-regressive or diffusion-based generative model [15, 69, 70] is learned to sample a 3D shape in latent from an image or text. However, generating a high-quality 3D shape from a 2D image or textual description is an ill-posed problem, and it usually requires more prior information for 3D shapes. In contrast, the distribution of the 3D shape space is significantly different from the 2D image or text space, and directly learning a probabilistic mapping function from the image or text to the 3D shape might reduce the quality, diversity, and semantic consistency with the conditional inputs. Prior research [69, 42] has endeavored to mitigate this concern through a coarse-to-fine approach, whereby the initial step involves generating coarse point clouds as an intermediary representation, followed by the regression of a neural field based on the point cloud.

Unlike the previous 3D shape representation, where the neural fields only characterize the geometric information of each 3D shape and capture the shape distribution by regularizing the shape latent with KL-divergence via VAE [13, 28, 71] or VQ-VAE [39, 70], we investigate a novel 3D shape representation that further brings the semantic information into the neural fields and designs a Shape-Image-Text-Aligned Variational Auto-Encoder (SITA-VAE). Specifically, it uses a perceiver-based transformer [62, 23] to encode each 3D shape into the shape embeddings and utilizes a contrastive learning loss to align the 3D shape embeddings with pre-aligned CLIP [48] image/text feature space from large-scale image-text pairs. After that, a transformer-based neural implicit decoder reconstructs the shape of latent embeddings to a neural occupancy or SDF field with a high-quality 3D shape. With the help of the aligned 3D shape, image, and text space, which closes the domain gap between the shape latent space and the image/text space, we propose an Aligned Shape Latent Diffusion Model (ASLDM) with a UNet-like skip connection-based transformer architecture [52, 5] to learn a better probabilistic mapping from the image or text to the aligned shape latent space and thereby generate a higher-quality and more diverse 3D shape with more semantic consistency conforming to the conditional image or text inputs.

To summarize, we explore bridging the semantic information into 3D shape representation via aligning the 3D shape, 2D image, and text into a compatible space. The encoded shape latent embeddings could also be decoded to a neural field that preserves high-quality details of a 3D shape. Based on the powerful aligned 3D shape, image, and text space, we propose an aligned shape latent diffusion model to generate a higher-quality 3D shape with more diversity when given the image or text input.

We perform extensive experiments on a standard 3D shape generation benchmark, ShapeNet [11], and a further collected 3D Cartoon Monster dataset with geometric details to validate the effectiveness of our proposed method. All codes will be publicly available.

## 2 Related Work

### 2.1 Neural 3D Shape Representation

Neural Fields have dominated the research of recent 3D shape representation, which predict the occupancy [37, 45], Sign Distance Function (SDF), density [43, 57], or feature vectors [10] of each 3D coordinate in the field via a neural network to preserve the high-fidelity of a specific 3D shape in a topology-free way. However, the vanilla neural field can only model a single 3D shape and cannot be generalized to other shapes. To this end, the researchers usually take additional latent codes, such as a global latent [43], a regular latent grid [45, 14], a set of latent points [69, 70], or latent embeddings [71, 25], which describe a particular shape along with each 3D coordinate to make the neural fields generalizable to other 3D shapes or scenes. Though current neural fields' 3D representation can characterize the low-level shape geometry information and preserve the high-fidelity shape details, bringing the high-level semantic information into the neural fields is still a relatively poorly studied problem. However, semantic neural fields are significant to downstream tasks, such as conditional 3D shape generations and 3D perception [22, 58].

### 2.2 Conditional 3D Shape Generation

**Optimization-based** approaches which employ a text-image matching loss function to optimize a 3D representation of the neural radiance field (NeRF). Dreamfields and AvatarCLIP [24, 21] adopt a pre-trained CLIP [48] model to measure the similarity between the rendering image and input text as the matching objective. On the other hand, DreamFusion [46] and Magic3D [29] utilize a powerful pre-trained diffusion-based text-to-image model as the optimization guidance and produce more complex and view-consistent results. However, per-scene optimization-based methods suffer from a low success rate and a long optimization time in hours to generate a high-quality 3D shape. However, they only require a pre-trained CLIP or text-to-image model and do not require any 3D data.

**Optimization-free** methods [33, 31, 32] are an alternative approach to conditional 3D shape generation that leverages paired texts/3D shapes or images/3D shapes to directly learn a conditional generative model from the text or image to the 3D shape representations. CLIP-Forge [56] employs an invertible normalizing flow model to learn a distribution transformation from the CLIP image/text embedding to the shape embedding. AutoSDF [39], ShapeFormer [68], and 3DILG [70] explore an auto-regressive model to learn a marginal distribution of the 3D shapes conditioned on images or texts and then sample a regular grid latent or irregular point latent shape embeddings from the conditions. In recent years, diffusion-based generative models have achieved tremendous success in text-to-image, video, and human motion generation. Several contemporaneous works, including SDFusion [13], Diffusion-SDF [28, 15], 3D-LDM [41], 3DShape2VecSet [71], and Shap-E [25], propose to learn a probabilistic mapping from the textual or visual inputs to the shape latent embeddings via a diffusion model. Since these approaches learn the prior information of the 3D shape data, they could improve the yield rate of high-quality shape generation. Moreover, there is no long-time optimization process, and the inference time is orders of magnitude faster than the optimization-based approaches. However, directly learning a conditional generative model to sample the 3D shape from the conditions might produce low-quality, less-diversity results due to the significant distribution gap between the shape and image/text spaces.

### 2.3 Contrastive Learning in 3D

Contrastive Language-Image Pre-training (CLIP) [48] has emerged as a fundamental model in 2D visual recognition tasks and cross-modal image synthesis by building the representation connection between vision and language within an aligned space. Recent works [12, 1, 53, 64] have extended the multi-modal contrastive learning paradigm to 3D. CrossPoint [2] learns the 3D-2D alignment to enhance the 3D point cloud understanding. PointCLIP [74] takes full advantage of the CLIP model pre-trained on large-scale image-text pairs and performs alignment between CLIP-encoded point cloud and 3D category texts to generalize the ability of 3D zero-shot and few-shot classification.

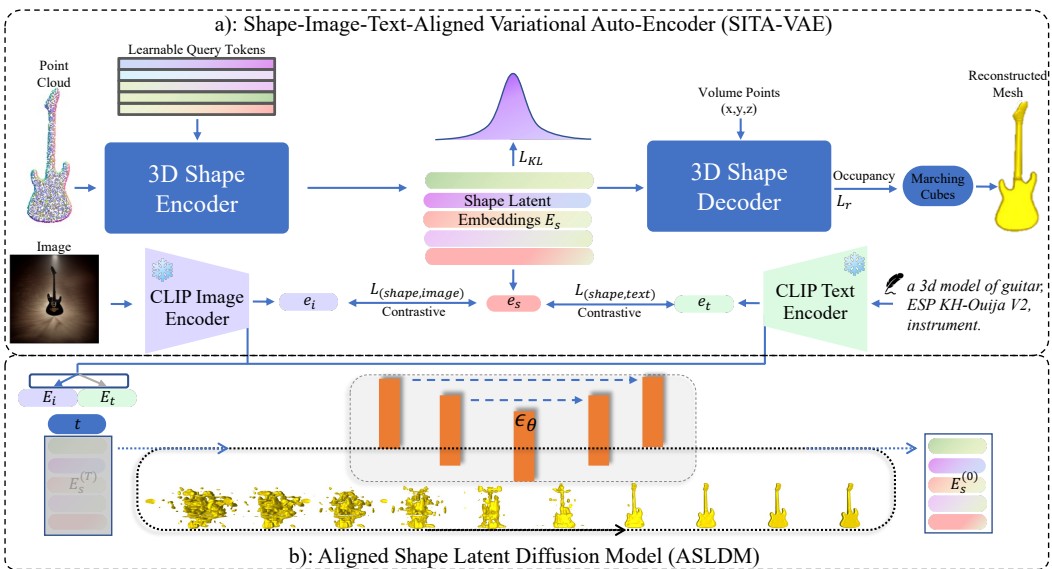

Figure 2: **Alignment-before-generation pipeline**. Our method contains two models: the Shape-Image-Text-Aligned Variational Auto-Encoder (SITA-VAE) and the Aligned Shape Latent Diffusion Model (ASLDM). The SITA-VAE consists of four modules: an image encoder, a text encoder, a 3D shape encoder, and a 3D shape decoder. Encoders encode inputs pair into an aligned space, and the 3D shape decoder reconstructs 3D shapes given embeddings from the aligned space. The ASLDM maps the image or text condition to the aligned shape latent space for sampling a high-quality 3D shape embedding, which is later reconstructed to high-fidelity 3D shapes by the 3D shape decoder.

ULIP [67], CLIP-goes-3D [16] and OpenShape [30] further learn a unified and aligned representation of images, texts, and 3D point clouds by pre-training with object triplets from the three modalities to improve 3D understanding. While most of these works focus on 3D recognition tasks, establishing the connection between 3D recognition and generation tasks remains an under-explored problem.

## 3 Our Approach

The direct learning of a probabilistic mapping from images or texts to 3D shapes is prone to produce inconsistent results due to the significant distribution gap between the 3D shapes and the 2D images and texts. To address this issue, we propose an alignment-before-generation solution for cross-modal 3D shape generation, as illustrated in Figure 2. Our approach involves two models: the Shape-Image-Text-Aligned Variational Auto-Encoder (SITA-VAE)(Section 3.1) and the Aligned Shape Latent Diffusion Model (ASLDM) (Section 3.2). The former model learns an alignment among the 3D shapes, images, and texts via contrastive learning and then reconstructs the shape embeddings back to the neural field. The latter model is based on the aligned space and is designed to learn a better conditional generative model from the images or texts to shape latent embeddings. By adopting this alignment-before-generation approach, we aim to overcome the challenges posed by the distribution gap and produce more consistent and high-quality results in cross-modal 3D shape generation.

### 3.1 Shape-Image-Text Aligned Variational Auto-Encoder

Our SITA-VAE contains four components: a pre-trained and fixed CLIP image encoder $\mathcal{E}_i$ and CLIP text encoder $\mathcal{E}_t$, a trainable 3D shape encoder $\mathcal{E}_s$ and neural field decoder $\mathcal{D}_s$. The CLIP image encoder and text encoder take 2D images $\mathbf{I} \in \mathbb{R}^{H \times W \times 3}$ and tokenized texts $\mathbf{T} \in \mathbb{R}^{L_t \times d_t}$ as input, and generate image tokens $\mathbf{E}_i \in \mathbb{R}^{(1+L_i) \times d}$ and text tokens $\mathbf{E}_t \in \mathbb{R}^{L_t \times d}$, where $(1 + L_i)$ and $L_t$ are the sequence length of image tokens $\mathbf{E}_i$ and text tokens $\mathbf{E}_t$. We take advantage of the pre-trained image encoder and text encoder from CLIP. These two encoders are trained on large-scale image-text pairs and robust enough to capture a well-aligned vision-language space, which will enrich the semantics of the 3D shape representation after multi-modal alignment via contrastive learning.

**3D shape encoder** aims to extract powerful feature representations to characterize each 3D shape effectively. To achieve this, we first sample point clouds $\mathbf{P} \in \mathbb{R}^{N \times (3+C)}$ from the surface of 3D shapes, where $N$ represents the number of points, and $C$ denotes additional point features such as normal or color. Next, we use a linear layer to project the concatenation of the Fourier positional encoded point clouds $\mathbf{P}$ to the 3D shape encoder input $\mathbf{X} \in \mathbb{R}^{N \times d}$. Drawing inspiration from previous transformer-based architectures for point cloud understanding [23], we build our 3D shape encoder on a perceiver-based transformer. Specifically, we use a cross-attention layer to inject the 3D shape information from the input $\mathbf{X}$ into a series of learnable query tokens $\mathbf{Q} \in \mathbb{R}^{(1+L_s) \times d}$, where $1 + L_s$ is the length of query tokens $\mathbf{Q}$, consisting of one global head token $\mathbf{Q}_g \in \mathbb{R}^{1 \times d}$ with high-level semantics and $L_s$ local tokens $\mathbf{Q}_l \in \mathbb{R}^{L \times d}$ containing low-level geometric structure information. Then, several self-attention blocks are used to iteratively improve the feature representation and obtain the final shape embeddings, $\mathbf{E}_s \in \mathbb{R}^{(1+L_s) \times d}$.

**Alignment among 3D shapes, images, and texts** plays a crucial role in SITA-VAE and the conditional generative models. Since the 3D data is the order of magnitudes smaller than the images and texts data, to learn a better-aligned shape among 3D shapes, images, and texts, we enforce the 3D shape encoder close to a pre-aligned vision-language space, which is pre-trained on a large-scale image-text pair with rich image and text representations by leveraging the contrastive learning strategy. Consider an input pair of 3D shapes $\mathbf{X}$, images $\mathbf{I}$ and tokenized texts $\mathbf{T}$. The triplet encoders generate the corresponding shape embedding $\mathbf{e}_s$, image embedding $\mathbf{e}_i$ and text-embedding $\mathbf{e}_t$ by projecting the extracted shape tokens $\mathbf{E}_s$, image tokens $\mathbf{E}_i$ and text tokens $\mathbf{E}_t$ as three vectors with the same dimension, which is expressed as: $\mathbf{e}_s = \mathcal{F}_s(\mathbf{E}_s), \mathbf{e}_i = \mathcal{F}_i(\mathbf{E}_i)$, and $\mathbf{e}_t = \mathcal{F}_t(\mathbf{E}_t)$, where $\mathcal{F}_s$ is a learnable shape embedding projector, image embedding projector $\mathcal{F}_i$ and text embedding projector $\mathcal{F}_t$ are pre-trained and frozen during training and inference. The contrastive loss is:

$$
\begin{aligned}
\mathcal{L}_{(shape,image)} &= -\frac{1}{2} \sum_{(j,k)} (\log \frac{\exp(\mathbf{e}_s^j \mathbf{e}_i^k)}{\sum_l \exp(\mathbf{e}_s^j \mathbf{e}_i^l)} + \log \frac{\exp(\mathbf{e}_s^j \mathbf{e}_i^k)}{\sum_l \exp(\mathbf{e}_s^l \mathbf{e}_i^k)}), \\
\mathcal{L}_{(shape,text)} &= -\frac{1}{2} \sum_{(j,k)} (\log \frac{\exp(\mathbf{e}_s^j \mathbf{e}_t^k)}{\sum_l \exp(\mathbf{e}_s^j \mathbf{e}_t^l)} + \log \frac{\exp(\mathbf{e}_s^j \mathbf{e}_t^k)}{\sum_l \exp(\mathbf{e}_s^l \mathbf{e}_t^k)}),
\end{aligned}
\tag{1}
$$

where $(j, k)$ indicates the positive pair in training batches, and since we utilize pre-trained encoders from CLIP, the model is free from constraint $\mathcal{L}_{(image,text)}$.

**3D shape decoder**, $\mathcal{D}_s$, takes the shape embeddings $\mathbf{E}_s$ as inputs to reconstruct the 3D neural field in a high quality. We use the KL divergence loss $\mathcal{L}_{KL}$ to facilitate the generative process to maintain the latent space as a continuous distribution. Besides, we leverage a projection layer to compress the latent from dimension $d$ to lower dimensions $d_0$ for a compact representation. Then, another projection layer is used to transform the sampled latent from dimension $d_0$ back to high dimension $d$ for reconstructing neural fields of 3D shapes. Like the encoder, our decoder model also builds on a transformer with the cross-attention mechanism. Given a query 3D point $\mathbf{x} \in \mathbb{R}^3$ in the field and its corresponding shape latent embeddings $\mathbf{E}_s$, the decoder computes cross attention iterative for predicting the occupancy of the query point $\mathcal{O}(x)$. The training loss is expressed as:

$$
\mathcal{L}_r = \mathbb{E}_{x \in \mathbb{R}^3}[\mathrm{BCE}(\mathcal{D}(\mathbf{x}|\mathbf{E_s}), \mathcal{O}(\mathbf{x}))],
\tag{2}
$$

where BCE is binary cross-entropy loss, and the total loss for training Shape-Image-Text Aligned Variational Auto-Encoder (SITA) is written as:

$$
\mathcal{L}_{SITA} = \lambda_c(\mathcal{L}_{(shape,image)} + \mathcal{L}_{(shape,text)}) + \mathcal{L}_r + \lambda_{KL}\mathcal{L}_{KL}.
\tag{3}
$$

### 3.2 Aligned Shape Latent Diffusion Model

After training the SITA-VAE, we obtain an alignment space among 3D shapes, images, and texts, as well as a 3D shape encoder and decoder that compress the 3D shape into low-dimensional shape latent embeddings and reconstruct shape latent embeddings to a neural field with high quality. Building on the success of the Latent Diffusion Model (LDM) [51] in the text-to-image generation, which strikes a balance between computational overhead and generation quality, we propose a shape latent diffusion model on the aligned space to learn a better probabilistic mapping from 2D images or texts to 3D shape latent embeddings. By leveraging the alignment space and the shape latent diffusion model, we can generate high-quality 3D shapes that better conform to the visual or textural conditional inputs.

Our Aligned Shape Latent Diffusion Model (ASLDM) builds on a UNet-like transformer [52, 62, 5], aim to fit a distribution of the shape latent embeddings, accompanied by an auto-encoder for encoding data samples into the latent space and reconstructing the data samples given the sampled latent. By learning in the latent space, the latent diffusion model is computationally efficient, and leveraging such a compact representation enables the model to fit the target distribution faster. Specifically, the model $\epsilon_\theta$ focuses on generating shape latent embeddings $\mathbf{E}_s$ conditioned on $\mathbf{C}$, which is represented by the CLIP image or text encoder. Following LDM [51], the objective is

$$\mathcal{L} = \mathbb{E}_{\mathbf{E}_s, \epsilon \sim \mathcal{N}(0,1), t}[\|\epsilon - \epsilon_\theta(\mathbf{E}_s^{(t)}, \mathbf{C}, t)\|_2^2], \tag{4}$$

where $t$ is uniformaly samppled from $\{1, ..., T\}$ and $\mathbf{E}_s^{(t)}$ is a noisy version of $\mathbf{E}_s^{(0)}$. During inference, sampling a Gaussian noise, the model gradually denoises the signal until reaching $\mathbf{E}_s^{(0)}$. Followed with classifier-free guidance (CFG) [20], we train our conditional latent diffusion model with classifier-free guidance. In the training phase, the condition $\mathbf{C}$ randomly converts to an empty set $\emptyset$ with a fixed probability $10\%$. Then, we perform the sampling with the linear combination of conditional and unconditional samples:

$$\epsilon_\theta(\mathbf{E}_s^{(t)}, \mathbf{C}, t) = \epsilon_\theta(\mathbf{E}_s^{(t)}, \emptyset, t) + \lambda(\epsilon_\theta(\mathbf{E}_s^{(t)}, \mathbf{C}, t) - \epsilon_\theta(\mathbf{E}_s^{(t)}, \emptyset, t)), \tag{5}$$

where $\lambda$ is the guidance scale for trading off the sampling fidelity and diversity.

## 4 Experiments

To validate the effectiveness of our proposed framework, we conducted extensive experiments. In this section, we provide implementation details of our model in Section 4.1. We also describe the data preparation process, including comparisons with baselines and metrics used in our evaluation, in Section 4.2. Of particular importance, we present quantitative comparison results to validate our model's generation ability. Additionally, we provide visual comparison results to illustrate the quality of the generative outputs in Section 4.3. Also, we conduct ablation studies in Section 4.4 to validate the effectiveness of training the generative model in the aligned space, the effectiveness of pre-trained vision-language models (VLM) on the SITA-VAE and the impact of learnable query embeddings.

### 4.1 Implementations

We implement our Shape-Image-Text-Aligned Variational Auto-Encoder (SITA-VAE) based on perceiver-based transformer architecture [23], where the 3D shape encoder consists of 1 cross-attention block and eight self-attention blocks. At the same time, the neural field decoder has 16 sefl-attention blocks with a final cross-attention block for the implicit neural field modeling. All attention modules are the transformer [62] style with multi-head attention mechanism (with 12 heads and 64 dimensions of each head), Layer Normalization (Pre-Norm) [3], Feed-Forward Network (with 3072 dimensions) [62] and GELU activation [17]. The learnable query embeddings are $\mathbf{E} \in \mathbb{R}^{513 \times 768}$ with one head-class token for multi-modal contrastive learning and left 512 shape tokens with a linear projection layer to the VAE space $\in \mathbb{R}^{512 \times 64}$ for the 3D shape reconstruction. Moreover, we employ pre-train encoders in the CLIP (*ViT-L-14*) [48] as our visual encoder and text encoder and freeze them during training and sampling. Besides, our aligned shape latent diffusion model (ASLDM) builds on a UNet-like transformer [52, 62, 5] consisting of 13 self-attention blocks with skip-connection by default. It contains 12 heads with 64 dimensions for each and 3076 dimensions in the Feed-Forward Network. Both models use an AdamW-based gradient decent optimizer [34] with a 1e-4 learning rate. Our framework is implemented with PyTorch [44], and we both train the SITA-VAE and ASLDM models with 8 Tesla V100 GPUs for around 5 days. We use the DDIM sampling scheduler [60] with 50 steps, which generates a high-quality 3D mesh within 10 seconds.

### 4.2 Datasets and Evaluation Metrics

**Dataset**. We use a standard benchmark, ShapeNet [11], to evaluate our model, which provides about 50K manufactured meshes in 55 categories. Each mesh has a category tag and corresponding texts, like fine-grained categories or brief descriptions given by the creator. We follow the train/val/test protocol with 3DILG [70]. We further collected 811 Cartoon Monster 3D shapes with detailed structures, with 615 shapes for training, 71 for validation, and 125 for testing, to evaluate the models' ability to generate complex 3D shapes. To prepare the triplet data (3D shape, image, text), we first

augment the provided texts in two ways. First, we string the shape tag and corresponding description in the format "a 3D model of (*shape tag*), in the style of (*description*)" or "a 3D model of (*shape tag*), (*description*)". Then, inspired by ULIP [67], we also leverage multiple templates containing 65 predefined phrases to provide more text information during training. As for the image data, we render each mesh under four camera poses, augmenting and improving the rendering diversity via the depth-condition-based ControlNet [73].

**Metrics**. We use the Intersection of Union (IoU) to reflect the accuracy of reconstructions. Then, we propose two new metrics for evaluating 3D shape generation methods. The first is a shape-image score (SI-S). We use a 3D shape encoder and image encoder to extract corresponding shape embedding and image embedding and compute the Cosine Similarity of these two modalities. Another is a shape-text score (ST-S), which computes the similarity between the generated 3D shape and the conditional text input in the aligned shape embedding and text embedding space. Both metrics evaluate the similarity between results and their corresponding conditions. Moreover, we use both the pre-trained ULIP [67] and our SITA to compute SI-S and ST-S, in terms of SI-S (ULIP), ST-S (ULIP), SI-S (SITA) and ST-S (SITA), respectively. Besides, we follow the metrics of P-IS and P-FID as introduced in Point-E [42] and use a pre-trained PointNet++ [47] to compute the point cloud analogous Inception Score [55] and FID [18] to evaluate the diversity and quality of the generated 3D shapes.

### 4.3   Experimental Comparision

**Baselines**. In the representation stage, we compare our method with Occ [37], ConvOcc [45], IF-Net [14], 3DILG [70] and 3DS2V [71] on reconstruction tasks to valid the ability of the model to recover a neural field given shape embeddings on the ShapeNet dataset [11]. For the conditional generation stage, we choose the baselines of two recent powerful 3D generation methods, 3DILG and 3DS2V. We first finetune their shape representation module on a mixture dataset of the ShapeNet and the 3D Cartoon Monster. Then, we retrain the text and image conditional generative models of 3DILG and 3DS2V with all the same protocols as ours.

|  | Overall | Selected | Table | Chair | Airplane | Car | Rifle | Lamp |
|---|---|---|---|---|---|---|---|---|
| OccNet [37] | 0.825 | 0.81 | 0.823 | 0.803 | 0.835 | 0.911 | 0.755 | 0.735 |
| ConvOccNet [45] | 0.888 | 0.873 | 0.847 | 0.856 | 0.881 | 0.921 | 0.871 | 0.859 |
| IF-Net [14] | 0.934 | 0.924 | 0.901 | 0.927 | 0.937 | 0.952 | 0.914 | 0.914 |
| 3DILG [70] | 0.950 | 0.948 | 0.963 | 0.95 | 0.952 | 0.961 | 0.938 | 0.926 |
| 3DS2V [71] | 0.955 | 0.955 | **0.965** | 0.957 | 0.962 | 0.966 | 0.947 | 0.931 |
| Ours | **0.966** | **0.964** | **0.965** | **0.966** | **0.966** | **0.969** | **0.967** | **0.95** |

Table 1: **Numerical results for reconstruction comparison on IoU(↑, a larger value is better)**. The results show that our model performs best in 55 categories. The results of selected categories further prove that our model could reconstruct each category faithfully.

|  | Image-Conditioned | | | | Text-Conditioned | | | |
|---|---|---|---|---|---|---|---|---|
|  | SI-S (ULIP)↑ | SI-S (SITA)↑ | P-FID↓ | P-IS↑ | ST-S (ULIP)↑ | ST-S (SITA)↑ | P-FID↓ | P-IS↑ |
| 3DILG | 9.134 | 11.703 | 4.592 | 12.247 | 10.293 | 6.878 | 10.283 | 12.921 |
| 3DS2V | 13.289 | 15.156 | 2.921 | 12.92 | 12.934 | 9.833 | 5.704 | 13.149 |
| Ours | **13.818** | **15.206** | **1.586** | **13.233** | **16.647** | **13.128** | **2.075** | **13.558** |

Table 2: **Numerical results for conditional generation comparison**. The results show that our model achieves the best generative performance. The SI-S and ST-S indicate that our model generates high-fidelity results by well-mapping the condition information to its related 3D shapes. Moreover, P-FID reflects that our model generates the most realistic 3D shapes, and P-IS indicates that the generated samples are diverse. ↑ means a larger value is better, and ↓ otherwise.

**Numerical Comparison**. We report the numerical results in Table 1 and Table 2. Table 1 shows that our model has the best reconstruction performance on 55 overall categories, surpassing the rest. Results of the selected category further prove that our model could faithfully reconstruct 3D shapes in each of the 55 categories. Table 2 reports the numerical results for conditional 3D shape generation. Our model achieves the best on all the SI-S and ST-S, indicating that it could map the information from the image or text to its corresponding 3D shape information for generating high-fidelity results. Moreover, the P-FID proves that our model could produce high-quality shape-tokens for generating realistic 3D shapes, and P-IS indicates the diversity of the samples. Specifically, the four left columns

show that our model surpasses the baselines on image-conditioned generation, proving that our model can better map visual information to 3D shapes. The four right columns validate the generative quality of text-conditioned generation. Since natural language, compared to the 2D image, usually provides limited and abstract information, learning a model to map text information to the 3D shape is challenging. However, benefiting from training on the aligned latent space, our model significantly improves text-conditioned generation, shown in the right of Table 2, which reflects that our model well-maps natural language information to 3D shapes and generates diverse and high-quality results.

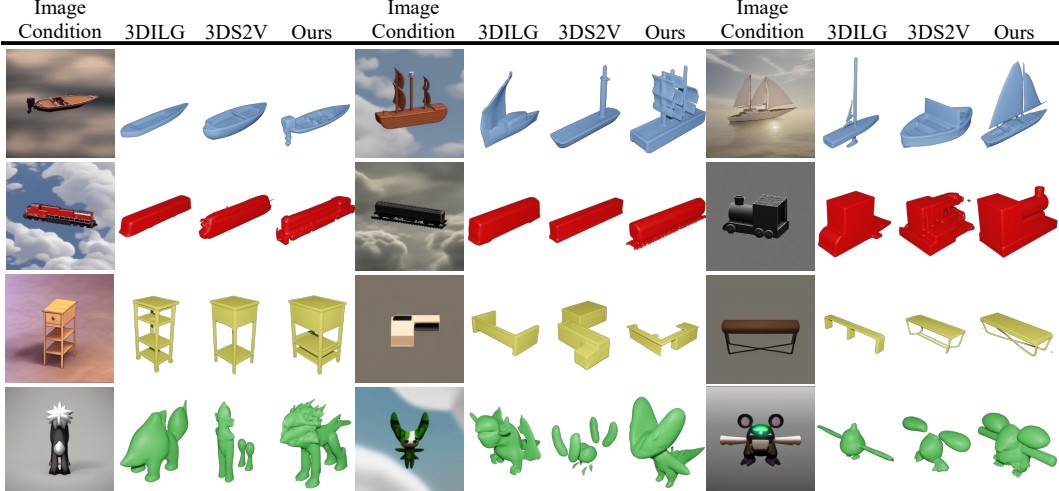

Figure 3: **Visual results for image-conditioned generation comparison**. The figure shows that 3DILG [70] generates over-smooth surfaces and lacks details of shapes, whereas 3DS2V [71] generates few details with noisy and discontinuous surfaces of shapes. In contrast to baselines, our method produces smooth surfaces and portrays shape details. Please zoom in for more visual details.

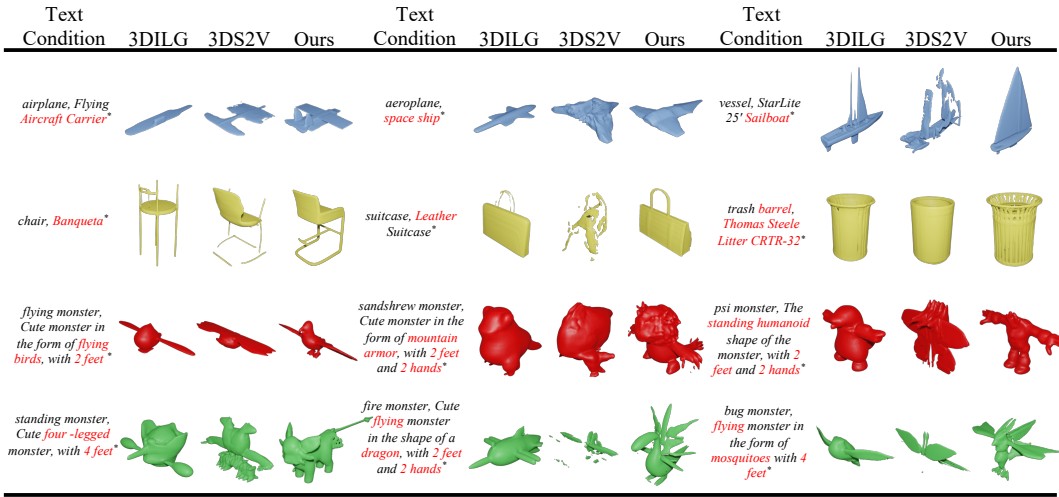

*denotes "a 3D model of …"

Figure 4: **Visual results for text-conditioned generation comparison**. In the first two rows, we test the model with abstract texts, and the result shows that only our model could generate a 3D shape that conforms to the target text with a smooth surface and fine details. The last two rows show the result given texts containing detailed descriptions, which further shows that our model could capture the global conditional information and the local information for generating high-fidelity 3D shapes. Keywords are highlighted in red; please zoom in for more visual details.

**Visual Comparison**. The visual comparisons of the image- and text-conditional 3D shape generations are illustrated in Figure 3 and Figure 4. Figure 3 shows that 3DILG [70] pays more attention to the global shape in the auto-regressive generation process, where its results lack depictions of details of

3D shapes. While 3DS2V [71] generates more details of 3D shapes, the results have discontinuous or noisy surfaces. Besides, both methods are unstable to generate a complete shape when the given conditions map to a complex object, fine machine, or rare monster. Figure 4 shows the visual comparison of text-conditional generation. In the upper-half rows, we show the results given simple and abstract concepts, while in the lower-half rows, we show the results given detailed texts like descriptions for deterministic parts of the target shape. Similar to the observation above, 3DILG [70] generates an over-smooth shape surface with fewer details, and 3DS2V [71] produces fewer details on the discontinuous object surface. Therefore, only our model produces correct shapes that conform to the given concepts or detailed descriptions with delicate details on smooth surfaces.

## 4.4 Ablation Studies and Analysis

We ablation study our model from three perspectives: the effectiveness of training generative model in the aligned space, the effectiveness of vision-language models (VLMs) on the SITA-VAE, and the impact of learnable query embeddings.

**The effectiveness of training generative model in the aligned space**. We perform a visual comparison for ablation study the effectiveness of training the generative model in the aligned space, as illustrated in the Figure 5. The uppers are sampled from the generative model that trains in the aligned space, while the lowers are sampled from the generative model that trains without aligned space. It proves that the uppers conform to the given text and the lower do not, which indicates that training the generative model in the aligned space leads to high-fidelity samples.

a 3d model of table, regulation table, tennis table, bench.     a 3d model of vintage car, 1938 Alfa Romeo 8C 2900B.

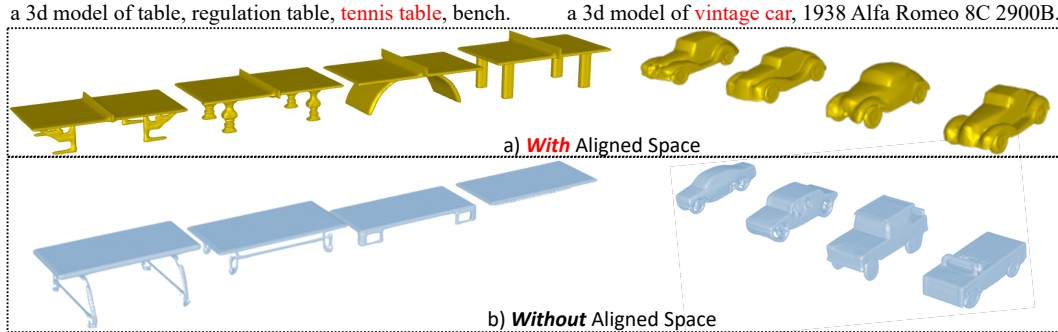

a) **_With_** Aligned Space

b) **_Without_** Aligned Space

Figure 5: **Ablation study the effectiveness of training generative model in the aligned space**. This figure illustrates visual comparisons for ablation studies on the effectiveness of training the generative model in the aligned space. Compared with the lower samples based on the conditional texts, the upper samples are closer to the conditions semantically, which indicates the effectiveness of the training generative model in the aligned space.

**The effectiveness of vision-language models**. Except for the well-known vision-language model (VLM) CLIP [48], we introduce another vision-language model (VLM) SLIP [40] for training the SITA-VAE for a comprehensive comparison. First, we evaluate the impact of the vision-language model on SITA-VAE's reconstruction ability, and the results are shown in Figure 6. It shows that our model composed with CLIP achieves the best performance. Then, we evaluate the vision-language model's impact on the ability to align multi-modal space. We select standard and zero-shot classification tasks to reflect the impact of the vision-language model. Note that the classification is performed by a feature matching operation, where we provide multiple 3D shapes and phrases to the SITA-VAE; it returns the similarity between 3D shapes to each phrase as classification results, which indicates that the more the multi-modal space is aligned, leading the higher classification accuracy. The results show that our model composed with CLIP achieves the best performance.

**The impact of the learnable query embeddings**. We ablation study learnable query embeddings with the same experiments as the above, and the results show that using 512 learnable query embeddings leads to the best performance on reconstructions and classifications.

## 4.5 Nearest Neighbor Analysis

In order to verify whether the model learns to generate results based on the given conditions or memorizes the whole training split of the dataset, we traverse the training set to retrieve the nearest

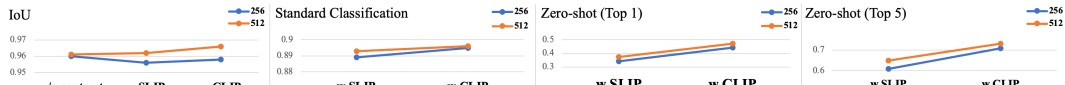

Figure 6: **Ablation study the effectiveness of vision-language models and the impact of learnable query embeddings**. This figure shows the ablation study on the effectiveness of the vision-language model and the impact of learnable query embeddings. According to the table, our model with CLIP and 512 learnable query embeddings achieves the best reconstruction and classification performance, indicating its ability to recover 3D shapes and align multi-modal space.

neighbors of the generated samples, as illustrated in Figure 7. Specifically, we exhibit three nearest neighbors for each sample for comparison. According to the visualization, the generated 3D shape differs from each retrieved nearest neighbor, demonstrating that our model hallucinates a new 3D shape rather than overfitting the training sets.

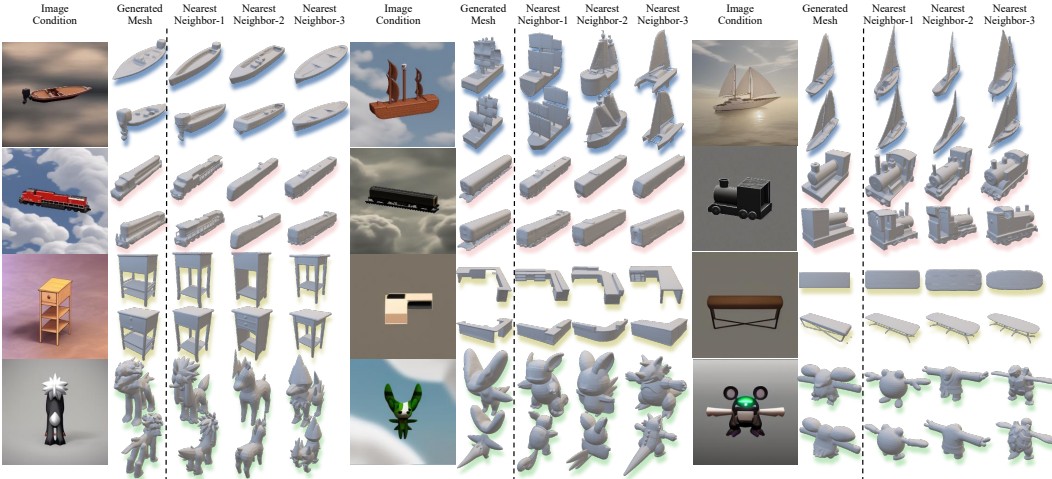

Figure 7: **Nearest Neighbor Analysis**. We traverse the whole training set to find three nearest neighbors for the generated 3D shapes, and the results reveal that our model could produce novel 3D shapes based on given images instead of memorizing specific ones.

## 5 Discussion and Conclusion

Though our method has achieved excellent results, it still has some limitations. First, our method needs the ground truth 3D shapes from training, while 3D data is usually an order of magnitude smaller than the 2D data. Learning the shape representation with a 3D shape-image-text aligned space from only 2D (multi-view) images via differentiable rendering is a promising direction. Furthermore, since we represent each 3D shape as an occupancy field, it needs to convert the 3D mesh into a watertight one, which will inevitably degrade the original quality of the 3D mesh.

In conclusion, we propose a novel framework for cross-modal 3D shape generation that involves aligning 3D shapes with 2D images and text. We introduce a new 3D shape representation that can reconstruct high-quality 3D shapes from latent embeddings and incorporate semantic information by aligning 3D shapes, 2D images, and text in a compatible space. This aligned space effectively closes the domain gap between the shape latent space and the image/text space, making it easier to learn a better probabilistic mapping from the image or text to the aligned shape latent space. As a result, our proposed method generates higher-quality and more diverse 3D shapes with greater semantic consistency that conform to the conditional image or text inputs.

**Acknowledgement** The work was supported by NSFC #61932020, #62172279, Program of Shanghai Academic Research Leader, and "Shuguang Program" supported by Shanghai Education Development Foundation and Shanghai Municipal Education Commission

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
