# OpenReview forum: "Michelangelo: Conditional 3D Shape Generation based on Shape-Image-Text Aligned Latent Representation"
_NeurIPS.cc/2023/Conference — NeurIPS 2023 poster_

### Official Review · Reviewer_73Gb · 2023-06-23

**Soundness:** 2 fair
**Presentation:** 3 good
**Contribution:** 2 fair
**Rating:** 5
**Confidence:** 3

**Summary:**

This paper proposes a conditional 3D generation method by pre-aligning features of different modalities when training a 3D AE. After that, a latent diffusion model is applied to generate latent vectors for 3D decoder conditioned on text/image. The results quantitatively and qualitatively shows that the proposed method is able to achieve impressive results.


**Strengths:**

1. The results show that this method is able to achieve multiple tasks, which is good and important for a generation-related paper.
2. The main figure is easy to understand and follow, making this paper easier to read.
3. The core idea in this paper looks original.


**Weaknesses:**

1. No qualitative results show the diversity of the proposed method. Such a result is important to a generation-related paper.
2. The concept of pre-alignment is not very convincing. Other conditional generation methods are also aligning text/image to the shape latent space by either pulling their representations together or learning diffusion models. Further discussion on how this concept is better than others should be added.


**Questions:**

1. If you have paired data for text-image-shape alignment training, why not just directly train a latent diffusion model conditioned on image/text to generate 3D objects like SDFusion[1]? In contrast, if you already aligned text/image/shape features in latent space, is it still necessary to use a heavy conditional generator such as latent diffusion to achieve your tasks? It feels like you only need one of these two techniques.
2. In your qualitative comparison, you mentioned that 3DS2V generates more details of 3D shapes and discontinuous surfaces and noisy results. However, such results are not seen in their paper, I wonder the reason why they fail to produce good results in Figure 2 and 3. Is it because you re-train their model on your dataset or because you directly use their pre-trained checkpoint or other reasons? More discussion and insight should be addressed.
3. You pre-aligned the text/image/shape feature first and use text/image feature to train a latent diffusion model to achieve conditional generation. However, as far as I know, if the input condition and the target feature are too similar when training the diffusion model, the diversity of the generation result would be limited and the purpose of using the diffusion model would disappear. How do you avoid such a situation from happening?
4. Please answer the problem mentioned in the weakness part.


**Limitations:**

yes

---

> ### Author Rebuttal · Authors · 2023-08-10
>
> We thank the reviewer for recognizing our paper's originality and importance to the community. Thanks for the appreciation for our visual results demonstrations and thoughtful comments. We will attempt to address your concerns in four aspects in the following.
>
> **Q1: Qualitative results of the diversity.**
>
> **A1:** We show qualitative results in Figure.1 of the rebuttal PDF, demonstrating the generated results' diversity. Moreover, we further illustrate the Nearest Neighbors from the training samples to verify that the trained model does not memorize the training samples in Figure.2 or the rebuttal PDF.
>
>
> **Q2: Motivations and effectiveness of alignment-before-generation approach.**
>
> **A2:** We recall our goals at the beginning of the response, developing a conditional 3D shape generative model to generate high-quality 3D shapes to conform to the given images/texts. And the model should work on all the categories of data, as many as we have, instead of a single category or a few.
>
> To achieve this, we start exploring training conditional 3D shape generative models on the whole ShapeNet, a large-scale 3D shape benchmark, several months ago. During the exploration, we find that even if a conditional 3D shape generative model trains successfully, the samples from the model do not conform to the corresponding conditions well, like the generated results of the text-to-image generative models. Besides, it is difficult for the model to map the condition to the corresponding 3D shapes for those categories with few training data.
>
> We observe that generating a 3D shape based on a 2D image or text is an ill-posed problem since conditions only provide partial information while the remaining information is missed, necessitating priors from the latent space of 3D shape distribution. However, the latent space of 3D shapes significantly differs from 2D images and texts and leads to difficulties for the model in mapping 2D information and textual information to valid priors in a pure 3D shapes latent space.
>
> However, most previous works tend to encode the 3D shapes into the latent space that contains only geometry information without considering the semantic information, and none of the existing works manage to train an image-/text-conditioned generative model on the whole ShapeNet. Therefore, we turn to represent 3D shapes in a latent space that captures both geometric and semantic information, which alleviates the difficulties for the generative models in mapping semantic information to 3D shape priors. Moreover, the success of DALLE-2, a generative model that first trains a prior model from an aligned CLIP text space to an aligned CLIP image space, inspires us to devise the alignment-before-generation approach.
>
> As a result, our model is the first that performs image- and text-conditioned 3D shape generation tasks on the whole ShapeNet categories, and the model exhibits generalization ability to unseen categories to some extent. Moreover, with the release of another large-scale dataset, Objaverse, and the incoming Objaverse-XL, we believe that our method can extend to and scale up to such a large-scale dataset (Currently, we have successfully scaled up our method in Objaverse, and the qualitative results in Figure.1 of the rebuttal PDF and the pre-trained model will be released.).
>
> **Q3: Relations between alignment and conditional generation.**
>
> **A3:** Thanks for the reviewer's interest in our alignment-before-generation approach. Alignment and conditional generative models could coexist, and our experiments reveal that alignment before generation enhances the generative model.
>
> * Contrastive pre-training cannot bring the generative ability to a model such as CLIP. It aims to train the model to capture multi-modal information in an aligned latent space. For SITA-VAE, contrastive learning constrains the model to capture geometric and semantic information in an aligned latent space. However, SITA-VAE can only reconstruct the neural field of a given 3D shape latent, necessitating a generative model for producing a 3D shape latent.
>
> * Our experimental results further prove the effectiveness of the training generative model on the aligned latent space. Besides, the recent foundation text-to-image generative model DALLE-2 also indicates that learning on the aligned latent space enhances the generation process. Specifically, DALLE-2 trains a prior network that maps text embeddings to image embeddings on the latent space of CLIP and a decoder network to generate images with given image embeddings. The generative ability of DALLE-2 also proves the effectiveness of training the generative models on the aligned latent space.
>
> **Q4: Clarification on the comparison with 3DS2V.**
>
> **A4:**  According to the paper, 3DS2V only trains image-/text-conditioned generation tasks on a subset of the ShapeNet. And no pre-trained checkpoints are available on the all-category ShapeNet at that time. To make a fair comparison, we re-train 3DS2V with the same CLIP condition and parameters as ours.  It has two failure cases: the first is that although the model successfully maps given conditions to high-quality 3D shape latent, it only decodes a high-quality 3D shape, which is not visually or semantically close to the given condition. Another is even worse since mapping 2D information to 3D shapes is an ill-posed problem. When the model fails to sample validated 3D latent codes, and results in a noisy shape by the decoder. Visual and numerical results prove our statements. We observe that the image-conditioned generated results from 3DS2V's paper in Figure 13 are not as good as the category-conditioned generated results in Figure 8. It reflects our observation.

---

> > ### Comment · Reviewer_73Gb · 2023-08-10
> > **Following comment**
> >
> > Thank the authors for the hard work! The feedback has addressed most of my concerns.
> >
> > I will raise my rating to 5.

---

> > > ### Author Response · Authors · 2023-08-22
> > > **Thanks for the constructive comments and positive feedback!**
> > >
> > > Thanks for recognizing our response. We are glad about the favorable assessment of our paper. We will rearrange the qualitative results to demonstrate the generated diversity in the revision.

---

### Official Review · Reviewer_XFVg · 2023-06-27

**Soundness:** 3 good
**Presentation:** 3 good
**Contribution:** 3 good
**Rating:** 6
**Confidence:** 5

**Summary:**

This work is about 3D shape generative model focused on image-conditioned and text-conditioned generation. The authors aligned the latent space of a shape autoencoder to CLIP's image encoder and text encoder. Then generative diffusion models are trained on the aligned latent space. This enables shape generation given image or text as conditional input. The authors showed some good results in both tasks.

**Strengths:**

The authors showed some good generation results for both the task of image-conditioned generation and text-conditioned generation. The writing is also clear and easy to follow.

**Weaknesses:**

1. The autoencoder network (Fig 1 a) is similar to the network used in [63]. Thus the performance boost shown in Table 1 seems to be because of CLIP. This should be emphasized or ablated somewhere in the main paper.
2. Following the above comment (Table 1), the authors only compared with "Learned queries" results from [63]. According to [63], another design "Point queries" achieved better results than both this work and "Learned queries".
3. It would be better if the authors can show some visualization comparisons of the autoencoding results.
4. L136, the length is (1+L_i) instead of L_i. This is explained in later sections (L217) but is still causing confusion.
5. Another difference with respect to [63] is, when training the diffusion models, this work used a unet-style transformer [4] instead of a simple stacked self attention network.
6. Some pieces of writing can still be improved. For example, when talking about a design or an equation, we should mention why we are doing this or discuss some insights behind this.

**Questions:**

See the weakness section.

**Limitations:**

This work proposed a method for image-conditioned and text-conditioned shape generation. The authors combined several components (3DS2V-style shape autoencoding network, clip and unet-style transformer denoising network). All these components are not proposed by this work which weakens the novelty of this paper. However, I still believe the authors delivered some good results in shape generation. I hope the authors can clarify my concerns mentioned in the weakness section.

---

> ### Author Rebuttal · Authors · 2023-08-10
>
> Thanks to reviewer XFVg for the positive feedback and insightful comments. Moreover, we are encouraged that the reviewer appreciates our results and enjoys reading the manuscript. We reply to the concerns in six aspects below. Furthermore, we will update Table 1 and related discussion in future revisions.
>
> **Q1: Insights and motivations.**
>
> **A1**: At the beginning of our response, we reiterate our goals, insights, and discovery.
> 1. We study generating general 3D shapes based on given 2D images or texts.
>
> 2. Even if a conditional 3D shape generative model trains successfully on the whole Shapenet dataset, the models' samples do not meet the given conditions enough. Besides, for those categories with few training samples, the model is hard to map the condition to the corresponding 3D shapes.
>
> 3. We observe that generating a 3D shape based on a 2D image or text is an ill-posed problem since conditions only provide partial information while the remaining information is missing, necessitating priors from the latent space of 3D shape distribution. It leads to difficulties for the model mapping 2D and textual information to valid priors in a pure 3D shapes latent space.
>
> 4. However, most previous works tend to encode the 3D shapes into the latent space that contains only geometry information without considering the semantic information, and none of the existing works manage to train an image-/text-conditioned generative model on the whole Shapenet dataset.
>
> 5. We turn to represent 3D shapes in a latent space that captures geometric and semantic information, alleviating the generative models' difficulties in mapping semantic information to 3D shape priors. Moreover, the success of DALLE-2, which learns on aligned CLIP latent space, inspires us to devise the alignment-before-generation approach.
>
> 6. As a result, our model is the first to perform image- and text-conditioned 3D shape generation tasks on the Shapenet dataset.
>
> **Q2: Ablation on CLIP.**
>
> **A2:** Thanks for the thoughtful comments. In the main paper, we only show an unremarkable chart in the first column in Figure.5 in the main texts to ablate the effectiveness of utilizing different Vision-Language models (VLMs). We extend the experiments to ablate the impact of VLMs, and the results are shown in the table.
>
> The last three rows in the table are three cases we used to train the VAE. Ours (w/o C) is a single VAE train without the contrastive loss, Ours (w SLIP) indicates to utilize the image- and text-encoder from SLIP, and Ours (w CLIP) is the SITA-VAE, where we employ CLIP's image- and text-encoder to train the VAE. The result showcases the effectiveness of the utilization of CLIP.
>
> |                    | Overall | Selected | Table  | Chair  | Airplane | Car    | Rifle  | Lamp   |
> |:-------------------|:-------:|:--------:|:------:|:------:|:--------:|:------:|:------:|:------:|
> | 3DS2V (PQ)         | 0.967   | 0.967    | 0.971  | 0.964  | 0.969    | 0.969  | 0.96   | 0.956  |
> | Ours (w/o Contrsat)| 0.961   | 0.958    | 0.958  | 0.962  | 0.961    | 0.968  | 0.952  | 0.945  |
> | Ours (w SLIP)      | 0.962   | 0.956    | 0.96   | 0.959  | 0.957    | 0.966  | 0.954  | 0.937  |
> | Ours (w CLIP)      | 0.966   | 0.964    | 0.965  | 0.966  | 0.966    | 0.969  | 0.967  | 0.95   |
>
> **Q3: Comparison with Point Queries in 3DS2V.**
>
> **A3:** Thanks for carefully figuring out the incomplete comparison in Table 1. We replenish Table 1 and report the results above.
>
> 1. Performance and analysis: The performance difference is due to the different sizes of the 3D shape latent. 3DS2V employs a deterministic autoencoder, while SITA-VAE contains a KL block that compresses the 3D shape latent. Therefore, the 3D shape decoder in SITA-VAE reconstructs 3D shapes with the low dimension 3D shape latent, leading to lower performance. Moreover, such results are similar to the conclusion from section 8.1 in 3DS2V, which shows that the compressed latent decreases performance.
>
> 2. Motivation for using learnable query embeddings: We design the learnable query embeddings for utilizing the local embedding setting from Perceiver and the global embedding setting from CLIP. Specifically, there embed (L_s + 1) learnable query embeddings in the 3D shape encoder. To learn a semantically and geometrically aligned 3D shape representation globally and locally via the learnable query embeddings, it contains a global head-class token and L_s shape tokens. Moreover, the learnable query embeddings and the cross-attention mechanism allow the model to handle significant inputs, potentially suitable for further scaling up.
>
> **Q4: Comparison of auto-encoding.**
>
> **A4:** We show the visual comparison of auto-encoding in Figure.4 in the rebuttal PDF.
>
> **Q5: Concern 5: Clarification for the notion in line 136.**
>
> **A5:** The image encoder in CLIP is a ViT-based architecture. Thus, the sequence in the encoder consists of one head-class token and L_i local tokens from the patch embeddings, and the sequence length of the last hidden layer in ViT is (1 + L_i). The following is the utilization of tokens.
>
> 1. When computing the multi-modal contrastive losses, only the head-class token will be projected into an image embedding in the training process of SITA-VAE.
>
> 2. In training generative models, we extract the sequence from the last hidden layer in the image encoder, consisting of one head-class token and L_i local tokens to better capture global and local information.
>
> **Q6: The architecture of the Diffusion Model.**
>
> **A6:** We implement the denoiser based on two architectures. A UNet-like transformer and a stacked attention block network, where each attention block contains a self-attention module and a cross-attention module.
> During experiments, we found that the UNet-like transformer denoiser converges with fewer iterations, as concluded in [4], and the training phase is stable. Therefore, we employ the UNet-like transformer architecture for the rest of the experiments.

---

> > ### Comment · Area_Chair_zit6 · 2023-08-20
> >
> > Dear reviewer,
> >
> > Please look over the author response and the other reviews and update your opinion.  Please ask the authors if you have additional questions before the end of the discussion period.

---

> > ### Comment · Reviewer_XFVg · 2023-08-21
> >
> > Thanks for the clarification. I will keep my positive rating.

---

> > > ### Author Response · Authors · 2023-08-22
> > > **Thanks for the valuable comments and positive feedback!**
> > >
> > > Thanks for recognizing our response. We are glad about the favorable assessment of our paper. We will rearrange the experimental comparisons and clarification of the notions in the revision.

---

### Official Review · Reviewer_rVoM · 2023-07-06

**Soundness:** 3 good
**Presentation:** 3 good
**Contribution:** 2 fair
**Rating:** 5
**Confidence:** 4

**Summary:**

This paper proposed a conditional generation model which aims to solve the alignment issue in image-to-shape or text-to-shape generation. The key idea is to learn a aligned representation among 3D shapes, images, and texts. To achieve that, the author proposes SITA-VAE with contrastive loss to force the shape's latents to be aligned with the pretrained vision-language model. After that, a LDM is trained to learn the diffusion process in the latent space. In the test time, it follows the previous works to use classifier-free guidance to perform conditional generation. The proposed method is evaluated on ShapeNet and Cartoon Monster 3D shapes.

**Strengths:**

* Propose to learn a aligned space for 3D shapes, images, and texts with 2D vision-language model. This essentially leverages the abundant 2D data into 3D modeling. The scarcity of paired data is one crucial reason why conditional 3D modeling does not perform as well as its 2D counterpart.

* The conditional generation results look great and align with the inputs when compared with the baselines.

* It is surprised that the aligned space does hurt the reconstruction results.

**Weaknesses:**

* Novelty is somewhat limited. The first stage model is mostly based on 3DILG and the multi-modality conditional generation has been explored in previous work such as SDFusion.

* Scalability is an issue as the model need paired 3D-image-text data to work.

* Some parts of the writing and figure are misleading. The choices of the metric is problematic in Table 2.

Please see the "Questions" for the details.

**Questions:**

1. The results in Fig. 2 seems a bit too sharp and I wonder if the model is memorizing the outputs. Can the author shows the nearest neighbor sample from the datasets?

2. In figure 1, the model is outputting occupancy but the final result is denoted as "mesh". I think the output should be occupancy and the mesh is obtained via marchingcube, right? Learning to generate mesh directly is very different from generating occupancies. The writing also use the word "mesh" and it is a bit confusing.

3. In Table 2, the use of SITA to evaluate the performance of conditional generation is unfair compared to other baselines since the LDM is trained on the space learned with SITA.

**Limitations:**

Authors have address the limitations.

---

> ### Author Rebuttal · Authors · 2023-08-10
>
> Thanks to reviewer rVoM for the thoughtful comments. We appreciate the reviewer's approval of our align-before-generation approach. Furthermore, we are encouraged that the reviewer recognize our visual demonstrations. We dedicated replies to the reviewers' comments and questions in five aspects below.
>
> **Q1: Novelty.**
>
> **A1**:In the beginning, we acknowledge that image-/text-conditioned 3D shape generation tasks exist. Still, please reiterate that our alignment-before-generation approach is the first to train an image-/text-conditioned 3D shape generative model on the whole Shapenet Dataset. Furthermore, we are working on scaling our model on a larger scale 3D benchmark, and the model exhibits good scalability in our experiments.
> 1. The difference between 3DILG and the first stage model, Shape-Image-Text-Aligned Variational Auto-Encoder (SITA-VAE).
>
>     a. First, 3DILG only encodes 3D shape information in their latent space. In contrast, our SITA-VAE aligns semantic information into the 3D shape latent space by leveraging contrastive learning;
>
>     b. The second difference is that 3DILG encodes the 3D shape as paired explicit center-coordinates and points features, which requires Farthest Points Sampling and k-nearest-neighbor clustering to handle the input. At the same time, our SITA-VAE utilizes the learnable query embedding to encode the 3D shapes implicitly, and our encoder could directly learn geometric and semantic information from the whole input point clouds;
>
>     c. The third is that 3DILG employs eight-layer multi-layer perceptrons (MLPs) to reweight explicit centers-coordinates and the output tokens from its decoder and predict the occupancy of query points, while the decoder in our SITA-VAE utilizes cascade attention modules to predict the occupancy of query points;
>
>     d. 3DILG trains in a VQ-VAE manner.
>
> 2. The difference between our Aligned Shape Latent Diffusion Model (ASLDM) and previous multi-modality conditional generation, such as SDFusion.
>
>     a.First, our ASLDM differs from the previous methods in that ASLDM learns on an aligned latent space. Still, previous conditional generative models almost train on a single modal space.
>
>     b.The second is the representation of 3D shape latents. SDFusion encodes 3D shapes as discrete latent voxels via VQ-VAE, which is potentially challenging to scale up to fit higher resolutions shapes or larger datasets because it achieves it by using higher-resolution discrete latent voxels or larger codebooks. In contrast, our ASLDM learns on the continuous shape latent with low dimensions, which are flexible. After successfully training on the whole Shapenet dataset, we further train the model on a larger dataset, and the results in Figure.1 in the rebuttal PDF prove our preliminary achievement.
>
> **Q2: Scalability.**
>
> **A2:** We acknowledge that 3D data is crucial for the scalability of conditional 3D shape generation tasks. Fortunately, another large-scale benchmark, Objaverse, which is ten times the entire Shapnet dataset, has been released. Since we have developed an autonomous pipeline (mentioned in the global response part) for producing shape-image-text triplet, described below, we could quickly start training on the new dataset. Moreover, we have some preliminary achievements, shown in Figure.1 in the rebuttal PDF.
>
> **Q3: Clarification on Table 2.**
>
> **A3:** Before explaining the content of Table 2, we clarify the typo in Table 2, as the repeated SI-S (ULIP) and SI-S (SITA) under 'Text-Conditioned' should be ST-S (ULIP) and ST-S (SITA). Thanks for the correction.
> We propose two new metrics to evaluate the conditional 3D shape generative models. The first is the shape-image score (SI-S), measuring the similarity between image conditions and generated 3D shapes. In particular, we use an image encoder to convert the image condition into an image embedding and a 3D shape encoder to transform the sampled 3D shape into a shape embedding. The SI-S is defined as the Cosine Similarity of the extracted shape and image embeddings. Another metric is the shape-text score (ST-S), which measures the similarity between text conditions and generated 3D shapes and implements a similar approach to SI-S. During the evaluation, although the 3D encoders in the SITA-VAE could extract 3D shape embeddings aligned to CLIP space, as the reviewer points out, only employing the 3D shape encoders from SITA-VAE to compare the generated results is unfair since ASLDM trains on the latent space of SITA-VAE. Therefore, for a comprehensive evaluation and comparison, we utilize an additional 3D shape encoder from ULIP, which pre-trains for aligning to the CLIP space. In detail, SI-S (SITA) and ST-S (SITA) indicate the score computed with the 3D encoder from SITA-VAE, and SI-S (ULIP) and ST-S (ULIP) indicate the score calculated with the 3D encoder from ULIP. The result in Table 2 proves that our model performs better under both ULIP and SITA-VAE.
>
> **Q4: Nearest Neighbor of the generated results from datasets.**
>
> **A4:** Please refer to the visual demos in Figure.3 in the rebuttal PDF, which illustrates the Top 3 nearest neighbors of the generated 3D shapes from the **training** dataset. The visual demos indicate that our model can generate novel shapes based on the learned 3D shapes in the training set rather than retrieve or memorize a specific shape.
>
> **Q5: Clarification on Marching Cubes.**
>
> **A5:** We apologize for the misleading due to our negligence while drawing the pipeline. The direct output from the 3D shape decoder is the occupancy of the query point, and the mesh is produced via conducting the marching cube algorithm on the occupancy of sampled volume points. Thanks for the rectification, and we will add a legend that indicates the marching cubes. Moreover, the abuse of 'mesh' and '3D neural shape' is somewhat confounded in our paper. In the revision, we will correct the inappropriate words.

---

> > ### Comment · Reviewer_rVoM · 2023-08-17
> >
> > Thank you for the thorough replies and the effort of additional experiments. Most of my concerns are resolved and I am happy to adjust the rating to 5.

---

> > > ### Author Response · Authors · 2023-08-22
> > > **Thanks for the insightful comments and positive feedback!**
> > >
> > > Thanks to reviewer for recognizing our response and additional experiments in the rebuttal PDF. We are glad about the favorable assessment of our paper. We will rearrange the nearest neighbor experiments and the clarification on Marching Cubes in the revision.

---

### Official Review · Reviewer_fo3T · 2023-07-07

**Soundness:** 3 good
**Presentation:** 4 excellent
**Contribution:** 3 good
**Rating:** 6
**Confidence:** 5

**Summary:**

This paper proposed a VAE-based text-to-3D shape generation method. The authors designed an alignment-before-generation approach to narrow the gap between 3D shapes and the 2D or text condition. They first train a Shape-Image-Text-Aligned Variational Auto-Encoder to align the representations between the 3D shapes and the 2D or text inputs. Then, the use latent diffusion model to denoise the shape embeddings to match the conditions. The overall presentation is good and the experiments are extensive.

**Strengths:**

1. The narratives are good and sound.
2. The SITA-VAE helps to align the representations between the three modalities.
3. The generation only requires some denoising steps on the latent embeddings.
4. The experiments are extensive and the authors provide sufficient visual demos and show the effectiveness of the method.

**Weaknesses:**

1. It seems that shape-image-text align training requires a lot of paired shape-image-text data, which could be a huge challenge in the generalization to new categories.
2. The experimental implementation is not clear enough and could be improved.

**Questions:**

1. Since this method is trained with ground truth 3D data, which limited its generalization ability, would it be better to show some failure cases when the condition fails to lie in the aligned representation space?

**Limitations:**

The author mentioned the limitation of the requirement of the 3D training data, which could be a huge problem in the generalization of this method.

---

> ### Author Rebuttal · Authors · 2023-08-10
>
> Thanks to reviewer fo3T for the positive feedback and thoughtful comments, and we are encouraged that the reviewer recognizes our effort on visual demonstrations and enjoys reading the manuscript. In the following, we reply to the individual concerns in three aspects.
>
> **Q1: What if the condition fails to lie in the aligned representation space?**
>
> **A1:**
>
> 1. We additionally show results that conditions on the MVImgNet images contain different image styles and some unseen categories in Figure.3 in the rebuttal PDF. The visual demos validate the generalization ability of our model.
>
> 2. Alignment-before-generation enhances the generalization ability of our model to some extent, as the CLIP latent space is a highly expressive and semantically rich space that captures both visual and textual information. By aligning the 3D space to the CLIP latent space, the model can utilize the semantic information captured in the CLIP space to close the semantic distance among the 3D latent. Benefiting from this, when the condition is unseen data to our training set, the model at least generates a semantically coherent 3D shape latent.
>
> 3. We employ the feature of the last layer from the CLIP encoder rather than the final embeddings of it. The last hidden layer tokens capture global and local information from the conditions, leading the model learning to match the conditions to the target 3D shape latent in terms of global semantic information and aspects of the local semantic parts. Therefore, when the global condition information fails to lie in the aligned representation space, some local parts always lie in the space, which facilitates the model's generalization ability.
>
> 4. We acknowledge that generalization is a significant challenge, highlighting the need for more efficient methods and larger-scale 3D datasets from the entire research community. In this work, we propose the alignment-before-generation approach, which enhances the generalization ability of our model. Additionally, the recently released Objaverse dataset is valuable for developing more effective 3D shape generation models. The continued efforts of the research community in this direction will lead to significant advancements in the field of 3D shape generation.
>
> **Q2: Shape-image-text align training requires a lot of paired shape-image-text data, which could be a huge challenge in generalizing to new categories.**
>
> **A2:**
> 1. **Data preparation:** Please refer to global response.
>
> 2. **Generalization to new categories:** Our model is the first that performs image-/text-conditioned generation on the whole Shapenet dataset and shows potent generative ability on some common categories after training on the aligned latent space from the entire Shapenet dataset. Moreover, benefiting from the autonomous data pre-processing procedure, we could quickly scale the training data triplet on the recent 3D shape benchmark. Moreover, we find it practicable to fine-tune the model on an enlarged dataset containing the Shapenet dataset, Objaverse, and some other released 3D datasets. We also show some visual results in Figure.1 in the rebuttal  PDF. Our model shows vast potential to scale up as a foundation model for conditional 3D shape generation tasks.
>
> **Q3: Experimental Implementations.**
>
> **A3:** Since we present **Model Implementations** in *global response* and generations comparison details in **Q3: Clarification on Table 2.** of response to *Reviewer rVoM*. We present the **ablation study** supplementary.
>
> We ablation study our method in several aspects.
> 1. The effectiveness of training the generative model in the aligned space. Figure.4 from the main text compares the results of two generative models under the same conditions. Our ASLDM produced the upper results, and the lower results produced by the generative model that trains in a single 3D shape latent space.
> 2. The effectiveness of the vision-language model (VLM). We ablation study the efficacy of the VLM on four tasks and two VLM models, SLIP and CLIP. We compare three VAEs, training without contrastive loss, training with an image-encoder and text-encoder from SLIP, and training with an image-encoder and text-encoder from CLIP.
>
>     a. The first is reconstruction. We use 3D shape encoders from three VAEs to conduct the shape reconstruction tasks on the testing set of Shapenet.
>
>     b. The second is Standard Classification. We use the 3D shape encoders and text encoders from three VAEs to perform a classification task on the testing set of Shapenet. As there are 55 categories in Shapenet, we set 55 texts with a template: a 3D model of (*), where * indicates a category. After computing the Cosine Similarity between a given 3D shape and the 55 texts, we regard the category with the highest Cosine Similarity as the classification results.
>
>     c. The third and the fourth are Zero-shot Classifications, similar to the second task, but evaluating the testing set of ModelNet, which contains 40 categories. Specifically, we compute a Top 1 accuracy and Top 5 accuracy. When calculating the Top 1 accuracy, we only record the highest Cosine Similarity with the category corresponding to the 3D shapes as successful classification. Moreover, it is correct to compute the Top 5 accuracy if the category conforms to the 3D shape that comes in the Top 5 Cosine Similarity.

---

> > ### Comment · Area_Chair_zit6 · 2023-08-20
> >
> > Dear reviewer,
> >
> > Please look over the author response and the other reviews and update your opinion.  Please ask the authors if you have additional questions before the end of the discussion period.

---

> > ### Comment · Reviewer_fo3T · 2023-08-20
> >
> > Thank you for the detailed reply. I will keep my rating.

---

> > > ### Author Response · Authors · 2023-08-22
> > > **Thanks for the thoughtful comments and positive feedback!**
> > >
> > > Thanks for recognizing our response. We are glad about the favorable assessment of our paper and will rearrange the implementation details in the revision.

---

### Author Rebuttal · Authors · 2023-08-10

# To All Reviewers:

We express our gratitude and appreciation to all the reviewers contributing to the review process. The reviewers, fo3T, rVoM, XFVg, and 73Gb, have commended the paper for:
1. well-written presentation.
2. Good visual demos.
3. Solid technical foundation.

We are also thankful for their insightful comments, which have provided valuable feedback to enhance our work and generate promising ideas for future research. We have endeavored to respond to the reviewers' queries and remarks with utmost clarity and detail in our individual replies. If the paper is accepted, we will restructure the final camera-ready version and incorporate more discussion into the main text.

**Motivation and novelty (Reviewer fo3T, rVoM, XFVg).**

We present a novel alignment-before-generation approach for conditional 3D shape generation tasks, showcasing the shape-image-text-aligned space's effectiveness for cross-modality 3D shape generation.
1. Compared to the previous 3D shape representation methods characterizing the geometric information of each 3D shape only, we investigate a 3D shape representation that further brings the semantic information into the neural fields. Meanwhile, the 3D shape representation is decodable.
2. Compared to the previous 3D shape generation methods training on the partial Shapenet dataset, the proposed Aligned Shape Latent Diffusion Model (ASLDM) successfully trains on the composition of the whole Shapenet dataset and 3D Cartoon Monster dataset based on the proposed 3D shape representation.

**What does this work achieve:**

The contribution of this work lies in training an image-/text-conditioned 3D shape generative model on the composition of the whole Shapenet and 3D Cartoon Monster dataset, which is the first of its kind. The generated results conform to the given conditions, demonstrating the effectiveness of the proposed approach.

**What does this paper propose:**

This paper proposes an alignment-before-generation approach to tackle conditional generative models' challenges in producing consistent 3D shapes with the given conditions.

**What does this project mean:**

This project reveals a fundamental but systematic pipeline of 3D shape generation tasks. The proposed alignment-before-generation approach enhances and stabilizes the training process of the generative model, and the introduced scalable data pre-processing scheme manages to create Shape-Image-Text triplets based on existing and potential large-scale 3D shape benchmarks.

**What does the current ASLDM differ from that at the paper submission:**

Benefiting from the release of more extensive scale benchmarks, we scaled up the ASLDM, resulting in improved performance.

**Data preparation:**

During the data pre-processing, we build shape-image-text triplets to exploit large-scale models' power. We especially employ the depth-condition-based ControlNet to render images under different views. We design some render prompts for rendering 3D shapes into corresponding images via its different views' depth images. The images and render prompts are then collected as the paired images and texts. Moreover, the texts are collected in three manners:
1. The category and descriptions of the shape from the Shapenet dataset.
2. The render prompts mentioned above closely pair with rendered images.
3. Followed by the text template in recent works, ULIP.

**Model Implementation**:

1. Shape-Image-Text-Aligned Variational Auto-Encoder (SITA-VAE).

    a. SITA-VAE implements perceiver-based transformer architecture. The 3D shape encoder consists of 1 cross-attention block and eight self-attention blocks. All the attention modules consist of multi-head attention (with 12 heads and 64 dimensions of each head), layer normalization (Pre-Norm), Feed-Forward Network (with 3072 dimensions), and GELU activation.

    b. The learnable query embeddings (with 513 tokens and 768 dimensions of each token) consist of one head-class token and 512 shape tokens. The shape tokens will be projected into VAE space (with 512 tokens and 64 dimensions of each token ) after processing by the 3D shape encoder, and another projection layer transforms the sampled latent back (with 768 dimensions) for the 3D shape reconstruction. We compute the multi-modal contrastive losses on the head-class token and the KL-divergence loss on the shape tokens.

    c. The 3D shape decoder (neural field decoder) has 16 self-attention blocks with a final cross-attention block for the implicit neural field modeling. All the attention modules consist of multi-head attention (with 12 heads and 64 dimensions of each head), layer normalization (Pre-Norm), Feed-Forward Network (with 3072 dimensions), and GELU activation. Given a query 3D point and its corresponding shape tokens, the decoder computes self-attention iterative and cross-attention for predicting the occupancy of the query point. We compute the binary cross-entropy loss on the result and ground-truth.

    d. Image-/text-encoder from CLIP (ViT-L-14) in SITA-VAE are frozen during training and inference.

    e. During the training phase, we follow 3DILG, which normalizes all mesh into [-1,1] first and then separately and equally samples volume points (randomly sampled) and near-surface points with ground-truth inside/outside labels from the watertight mesh.

2. Aligned Shape Latent Diffusion Model (ASLDM)

    a. ASLDM builds on a UNet-like transformer architecture. It consists of 13 self-attention blocks with skip-connection between shallow and deep layers. All the attention modules consist of multi-head attention (with 12 heads and 64 dimensions of each head), Layer Normalization (Pre-Norm), and Feed-Forward Network (with 3072 dimensions).

    b. The ASLDM trains with the MSE loss and . The training diffusion steps are 1000, and beta \in [0.00085,0.012] with a scaled linear scheduler.

Both module uses an AdamW-based gradient decent optimizer with a 1e-4 learning rate.

---

### Decision · Program_Chairs · 2023-09-21

**Decision:**

Accept (poster)

**Comment:**

This work proposes using an aligned shape-image-text with a variational auto-encoder for conditioned shape generation from either text or image.  The aligned latent space for shape-image-text is trained to align shape embddings from a perceiver-based transformer encoder with CLIP image and text embeddings using contrastive losses.  At the same time, a transformer-based decoder is trained to reconstract the shape occupancy.  A diffusion model is trained to probabilistically map text and image embeddings to the shape space.

Reviewers are positive on this work, noting that the generation results look better than baseline methods.

The authors are encouraged to update their paper with feedback from the reviewers, and incorporate clarification and additional results from the rebuttal including:
1. Add discussion of limitations and generalization to new categories
2. Incorporate additional visualizations, ablations, and other clarifications
3. Improve writing to clarify deisng decisions, relation to prior work, and architecture details
4. Inclusion of relevant related work for Shape-Image-Text aligned representations and text to shape generation such as:

   Work on shape-image-text aligned spaces:
   - OpenShape: Scaling Up 3D Shape Representation Towards Open-World Understanding [Liu et al. 2023]
   - Text2Shape: Generating Shapes from Natural Language by Learning Joint Embeddings [Chen et al. ACCV 2018]
   - ShapeGlot: Learning Language for Shape Differentiation [Achilioptas et al. ICCV 2019]
   - TriCoLo: Trimodal Contrastive Loss for Fine-grained Text to Shape Retrieval [Ruan et al. 2022]
   - MXM-CLR: A Unified Framework for Contrastive Learning of Multifold Cross-Modal Representations [Wang et al. 2023].

   Work on text to shape generation
   - Towards implicit text-guided 3d shape generation, Liu et al. CVPR 2022
   - ISS: Image as Stepping Stone for Text-Guided 3D Shape Generation, Liu et al. ICLR 2023

5. Proofreading pass to fix typos (L115: UNIP => ULIP, L310: Disscusion => Discussion)